# Value Propagation Networks

**Nantas Nardelli** [*]
University of Oxford
nantas@robots.ox.ac.uk

**Gabriel Synnaeve**
Facebook AI Research
gab@fb.com

**Zeming Lin**
Facebook AI Research
zlin@fb.com

**Pushmeet Kohli** [†]
Google DeepMind
pushmeet@google.com

**Philip H. S. Torr**
University of Oxford
philip.torr@eng.ox.ac.uk

**Nicolas Usunier**
Facebook AI Research
usunier@fb.com

## Abstract

We present Value Propagation (VProp), a set of parameter-efficient differentiable planning modules built on Value Iteration which can successfully be trained using reinforcement learning to solve unseen tasks, has the capability to generalize to larger map sizes, and can learn to navigate in dynamic environments. We show that the modules enable learning to plan when the environment also includes stochastic elements, providing a cost-efficient learning system to build low-level size-invariant planners for a variety of interactive navigation problems. We evaluate on static and dynamic configurations of MazeBase grid-worlds, with randomly generated environments of several different sizes, and on a StarCraft navigation scenario, with more complex dynamics, and pixels as input.

## 1 Introduction

Planning is a key component for artificial agents in a variety of domains. However, a limit of classical planning algorithms is that one needs to know how to search for an optimal – or at least reasonable – solution for each instantiation of every possible type of plan. As the environment dynamics and states complexity increase, this makes writing planners difficult, cumbersome, or simply entirely impractical. This is among the reasons why "learning to plan" has been an active research area to address these shortcomings (Russell et al., 1995; Kaelbling et al., 1996). To be useful in practice we propose that methods that enable to learn planners should have at least two properties: they should be *traces free*, i.e. not require traces from an optimal planner, and they should *generalize*, i.e. learn planners that are able to function on plans of the same type but of unseen instance and/or planning horizons.

In Reinforcement Learning (RL), learning to plan can be framed as the problem of finding a policy that maximises the expected return from the environment, where such policy is a greedy function that selects actions that will visit states with a higher value for the agent. This in turns shifts the problem to the one of obtaining good estimates of state values. One of the most commonly used algorithms to solve this problem is Value Iteration (VI), which estimates the state values by collecting and propagating the observed rewards until a fixed point is reached. A policy – or a plan – can then be constructed by rolling out the obtained value function on the desired state-action pairs.

When the environment can be represented as an occupancy map, a 2D grid, it is possible to approximate this planning algorithm using a deep convolutional neural network (CNN) to propagate the rewards on the grid cells. This enables one to differentiate directly through the planner steps and perform end-to-end learning of the value function. Tamar et al. (2016) train such models – Value Iteration Networks (VIN) – with a supervised loss on the trace from a search/planning algorithm, with the goal to find the parameters that can solve the shortest path task in such environments by iteratively learning the value function using the

---

[*]Work was partly done while at Facebook AI Research.
[†]Work done while at Microsoft Research.

convnet. However, this baseline requires good target value estimates, violating our wished trace free property, and limiting its usage in interactive, dynamic settings. Furthermore, it doesn't take advantage of the model structure to generalise to harder instances of the task.

In this work we extend the formalization used in VIN to more accurately represent the structure of grid-world-like scenarios, enabling Value Iteration network modules to be naturally used within the reinforcement learning framework beyond the scope of the initial work, while also removing some of the limitations and underlying assumptions constraining the original architecture. We show that our models can not only learn to plan and navigate in dynamic environments, but that their hierarchical structure provides a way to generalize to navigation tasks where the required planning horizon and the size of the map are much larger than the ones seen at training time. Our main contributions include: (1) introducing VProp and MVProp, network planning modules which successfully learn to solve pathfinding tasks via reinforcement learning using minimal parametrization, (2) demonstrating the ability to generalize to large unseen maps when training exclusively on much smaller ones, and (3) showing that our modules can learn to plan in environments with more complex dynamics than a static grid world, both in terms of transition function and observation complexity.

## 1.1 Related work

Model-based planning with end-to-end architectures has recently shown promising results on a variety of tasks and environments, often using Deep Reinforcement Learning as the algorithmic framework (Silver et al., 2016; Oh et al., 2017; Weber et al., 2017; Groshev et al., 2017; Farquhar et al., 2017; Serban et al., 2018; Srinivas et al., 2018). 3D and 2D navigation tasks have also been tackled within the RL framework (Mirowski et al., 2016), with methods in some cases building and conditioning on 2D occupancy maps to aid the process of localization and feature grounding (Bhatti et al., 2016; Zhang et al., 2017; Banino et al., 2018).

Other work has furthermore explored the usage of VIN-like architectures for navigation problems: Niu et al. (2017) present a generalization of VIN able to learn modules on more generic graph structures by employing a graph convolutional operator to convolve through each node of the graph. Rehder et al. (2017) demonstrate a method for multi-agent planning in a cooperative setting by training multiple VI modules and composing them into one network, while also adding an orientation state channel to simulate non-holonomic constraints often found in mobile robotics. Gupta et al. (2017) and Khan et al. (2017) propose to tackle partially observable settings by constructing hierarchical planners that use VI modules in a multi-scale fashion to generate plans and condition the model's belief state.

## 2 Background

We consider the control of an agent in a "grid world" environment, in which entities can interact with each other. The entities have some set of attributes, including a uniquely defined type, which describes how they interact with each other, the immediate rewards of such interactions, and how such interactions affect the next state of the world. The goal is to *learn to plan* through reinforcement learning, that is learning a policy trained on various configurations of the environment that can generalize to arbitrary other configurations of the environment, including larger environments, and ones with a larger number of entities. In the case of a standard navigation task, this boils down to learning a policy which, given an observation of the world, will output actions that take the agent to the goal as quickly as possible. An agent may observe such environments as 2D images of size $d_{\mathrm{x}} \times d_{\mathrm{y}}$, with $d_{\mathrm{pix}}$ input panes, which are then potentially passed through an embedding function $\Phi$ (such as a 2D convnet) to extract the entities and generates some local embedding based on their positions and features.

## 2.1 Reinforcement Learning

The problem of reinforcement learning is typically formulated in terms of computing optimal policies for a Markov Decision Problem (MDP) (Sutton and Barto, 1998). An MDP is

defined by the tuple $(S, A, T, R, \gamma)$, where S is a finite set of states, $A$ is the set of actions $a$ that the agent can take, $T : s \to a \to s'$ is a function describing the state-transition matrix, $R$ is a reward function, and $\gamma$ is a discount factor. In this setting, an optimal policy $\pi^*$ is a distribution over the state-action space that maximises in expectation the discounted sum of rewards $\sum_k \gamma^k r_k$, where $r_k$ is the single-step reward. A standard method to find the optimal policy $\pi : s \to a$ is to iteratively compute the value function, $Q^\pi(s, a)$, updating it based on rewards received from the environment (Watkins and Dayan, 1992). Using this framework, we can view learning to plan as a *structured prediction* of rewards with the planning algorithm Value Iteration (Bertsekas, 2012) as inference procedure. Beside value-based algorithms, there exist other types which are able to find optimal policies, such as *policy gradient* methods (Sutton et al., 1999), which directly regress to the policy function $\pi$ instead of approximating the value function. These methods however suffer from high variance estimates in environments that require many steps. Finally, a third type is represented by actor-critic algorithms, which combine the policy gradient methods' advantage of being able to compute the policy directly, with the low-variance performance estimation of value-based RL used as a more accurate feedback signal to the policy estimator (Konda and Tsitsiklis, 2000).

## 2.2 VALUE ITERATION MODULE

Tamar et al. (2016) motivate the *Value Iteration module* by observing that for problems such as navigation, and more generally pathfinding, value iteration can be unrolled as a graph convolutional network, where nodes are possible positions of the agent and edges represent possible transitions given the agent's actions. In the simpler case of 2D grids, the graph structure corresponds to a neighborhood in the 2D space, and the convolution structure is similar to a convolutional network that takes the entire 2D environment as input.

More precisely, let us denote by $s$ the current observation of the environment (e.g., a bird's-eye view of the 2D grid), $q^0$ the zero tensor of dimensions $(A, d_\mathrm{x}, d_\mathrm{y})$, where $d_\mathrm{x}, d_\mathrm{y}$ are the dimension of the 2D grid, and $A$ the number of actions of the agent. Then, the Value Iteration module is defined by an embedding function of the state $\Phi(s) \in \mathbb{R}^{d_\mathrm{rew} \times d_\mathrm{x} \times d_\mathrm{y}}$ (where $d_\mathrm{rew}$ depends on the model), a transition function $h$, and performs the following computations for steps $k = 1...K$ where $K$ is the depth of the VI module:

$$\forall (i, j) \in [\![ d_\mathrm{x} ]\!] \times [\![ d_\mathrm{y} ]\!], \quad v_{ij}^k = \max_{a=1..A} q_{a,i,j}^k,$$
$$q^k = h(\Phi(s), v^{k-1}).$$

Given the agent's position $(x_0, y_0)$ and the current observation of the environment $s$, the control policy $\pi$ is then defined by $\pi(s, (x_0, y_0)) = \mathrm{argmax}_{a=1..A} q_{a,x_0,y_0}^K$.

We can rewrite the transition function $h$ as a convolutional layer:

(a) $\quad \bar{r}_{i,j} = \Phi(s)_{i,j}$,

(b) $\quad v_{i,j}^0 = 0, \quad q_{a,i,j}^k = \sum_{(i',j') \in \mathcal{N}(i,j)} p_{a,i'-i,j'-j}^{(v)} * v_{i',j'}^{k-1} + p_{a,i'-i,j'-j}^{(r)} * \bar{r}_{i',j'}, \quad v_{i,j}^k = \max_a q_{a,i,j}^{k-1},$

where $\mathcal{N}(i,j)$ is the set of neighbors of the cell $(i,j)$ and the cell itself. In practice, $\Phi(s)$ has several output channels $d_\mathrm{rew}$ and a varying number of parameters when used within a VI module on different tasks. $p_{a,i'-i,j'-j}^{(v)} \in \mathbb{R}$ and $p_{a,i'-i,j'-j}^{(r)} \in \mathbb{R}$ instead purely represent the parameters of the VI module.

This model is appealing as a neural network architecture because the step to compute each $q_{a,i,j}^k$ is a convolutional layer, thus enabling computing $q_{a,i,j}^K$ with a convolutional network of depth $K$ with shared weights and as many output channels as actions. The computation of $v_{i,j}^k$ are then the non-linearities of the network, which correspond to a max-pooling in the dimension of the output channels.

To clarify the relationship with the original value iteration algorithm for grid worlds, let us denote by $R_{a,i,j,i',j'}$ the immediate reward when taking action $a$ at position $(i, j)$ with arrival position $(i', j')$, $P_{a,i,j,i',j'}$ the corresponding transition probabilities, and $\gamma$ the discount

factor. Then value iteration in a 2D grid can be generally written as:

$$\forall (i,j) \in [\![\, d_{\mathrm{x}} \,]\!] \times [\![\, d_{\mathrm{y}} \,]\!], \;\; V_{ij}^k = \max_{a=1..A} Q_{a,i,j}^k,$$

$$\forall (a,i,j) \in [\![\, A \,]\!] \times [\![\, d_{\mathrm{x}} \,]\!] \times [\![\, d_{\mathrm{y}} \,]\!], \;\; Q_{a,i,j}^k = \sum_{(i',j') \in \mathcal{N}(i,j)} P_{a,i,j,i',j'} \left( R_{a,i,j,i',j'} + \gamma V_{i,j}^{k-1} \right).$$

where $\mathcal{N}(i,j)$ is prior knowledge about the states that are accessible from $(i,j)$. The implementation of the value iteration module described above thus corresponds to a specific parameterization of the reward $R_{a,i,j,i',j'}$ as functions of the starting and arrival states. More importantly, the implementation of the value iteration module with a convolutional network means that the transition probabilities, represented by the parameters $p_{a,i'-i,j'-j}^{(v)}$, are *translation-invariant*.

## 3 Models

The value iteration modules is appealing from a computational point of view because they can be efficiently implemented as convolutional neural networks. They are also conceptually appealing for the design of neural planning architectures because they give a clear motivation and interpretation for sharing weights between layers, and provide guidance as to the necessary depth of the network: the depth should be sufficient for the reward signals to propagate from "goal" states to the agent, and thus be some function of the length of the shortest path.

While the parametrization with shared weights reduces sample complexity, this architecture is also amenable to an interesting form of generalization: learning to navigate in small environments (small $d_{\mathrm{x}}, d_{\mathrm{y}}$ for training), and generalize to larger instances (larger $d_{\mathrm{x}}, d_{\mathrm{y}}$). That is, from the analogy with value iteration it follows that generalization to larger instances should require deeper networks, which is possible because the weights are shared. Yet, the experiments in Tamar et al. (2016), as well as our experiments presented in Section 4, show that learning VI modules with reinforcement learning remains very challenging and doesn't naturally generalize as thought, suggesting that the sample complexity is still a major issue.

We propose in this section two alternative approaches to the value iteration modules described above, with the objective to provide a minimal parametrization for better sample complexity and generalization, while keeping the convolutional structure and the idea of a deep network with shared weights. We mostly revisit the two design choices made in the VI module: first, we drop the main assumption of VI modules that the transition probabilities are translation-invariant by making them a function of the state, and look for the minimal parametrization of these state-dependent transition probabilities suitable for grid worlds. Second, we propose more constrained parametrizations of the immediate reward function to increase sample efficiency using reinforcement learning, thus allowing to exploit the structure of the architecture to achieve generalization.

### 3.1 Value-Propagation Module

Let us observe that in the simplest version of a grid world the dynamics are deterministic and actions only consist of moving into an adjacent cell. The world model should account for blocking cells (e.g. terrain or obstacles), while the reward function should account for goal states. Furthermore – and very importantly when implementing value iteration for episodic tasks – one needs to account for terminal states, which can be represented by absorbing states, i.e. states for which the only possible transition is a loop to itself with reward 0 (Sutton and Barto, 1998).

We propose to take these elements into account in the following architecture, which we call *Value Propagation* (VProp). Similarly to a VI module, it is here implemented with deep

convolutional networks with weights shared across layers:

(a) $\qquad \bar{r}_{i,j}^{\text{in}}, \bar{r}_{i,j}^{\text{out}}, p_{i,j} = \Phi(s)_{i,j},$

(b) $\qquad v_{i,j}^0 = 0, \quad v_{i,j}^{(k)} = \max\left(v_{i,j}^{(k-1)}, \max_{(i',j')\in\mathcal{N}(i,j)}\left(p_{i,j}v_{i',j'}^{(k-1)} + \bar{r}_{i',j'}^{\text{in}} - \bar{r}_{i,j}^{\text{out}}\right)\right),$

(c) $\qquad \pi(s,(i_0,j_0)) = \underset{i',j'\in\mathcal{N}(i_0,j_0)}{\operatorname{argmax}} \, v_{i',j'}^K.$

In VProp, all parameters are in the embedding function $\Phi$ – the iteration layers do not have any additional parameters. This embedding function has two types of output for each position $(i,j)$. The first type is a vector $(\bar{r}_{i,j}^{\text{in}}, \bar{r}_{i,j}^{\text{out}})$, which is used to model the full reward function $R_{a,i,j,i',j'}$ as $\bar{r}_{i',j'}^{\text{in}} - \bar{r}_{i,j}^{\text{out}}$. This is done to correctly deal with absorbing states: $\bar{r}^{\text{in}}$ intuitively models the reward obtained when the agent enters position $i,j$ (typically high for a goal state, negative for positions we should avoid), and $\bar{r}^{\text{out}}$ is the cost of going out of a position. Absorbing states can then be represented by setting $\bar{r}_{i,j}^{\text{in}} = \bar{r}_{i,j}^{\text{out}}$.

The second output given by $\Phi$ is a single value $p_{i,j}$, which represent a *propagation* parameter associated to the position, or simply a state-dependent discount factor (White, 2016): $p_{i,j} \approx 1$ means that the neighboring values $v_{i',j'}$ propagate through $i,j$, while $p_{i,j} \approx 0$ means that position $i,j$ is blocking, which typically arises for cells containing obstacles. In our implementations, all $\bar{r}^{\text{in}}$, $\bar{r}^{\text{out}}$, $p$ are kept in $[0,1]$ using a sigmoid activation function.

VProp corresponds to the value iteration algorithm in which the dynamics are deterministic and there is a one-to-one mapping between actions and adjacent cells. This mapping is assumed in equation *(c)*, in which the output is a position rather than an action. In practice, since $(i',j')$ is an adjacent cell, it is trivial to know which action needs be taken to get to that cell. However, in case the mapping is unknown, one can use $\pi(s) = F\left([v_{i',j'}^{(K)}]_{(i',j')\in\mathcal{N}(i_0,j_0)}\right)$ for some $F$ that takes the neighborhood of the agent as input and performs the mapping from the propagated values to actual actions.

It's important to stress that this particular architecture was designed for environments that can be represented and behave like 2D grid structures (e.g. robotic navigation), however the formulation can be easily extended to more general graph structures. More details on this can be found in Appendix A.

## 3.2 Max-Propagation Module

One of the main difficulties we observed with both the VI module and VProp when generalizing to larger environments is that obstacles/blocking cells on a fixed size grid can be represented in two different ways: either with a small value of propagation or with a large output reward. In practice, a network trained on grids of bounded sizes may learn to any valid configuration, but a configuration based on negative rewards and high propagation does not generalize to larger environments: a negative reward for going through an obstacle is not, in general, sufficient to compensate for the length of a correct path that needs to get around the obstacle. When the environment has fixed size, the maximum length of going around of an obstacle is known, and the reward can be set accordingly. But as the environment increases in size, the cost of getting around the obstacle increases and is not well represented by the negative reward anymore.

To overcome this difficulty, we propose an alternative formulation, the Max-Propagation module (MVProp), in which only positive rewards are propagated. This implies that the only way to represent blocking paths is by not propagating rewards – negative rewards are not a solution anymore. The MVProp module is defined as follows:

(a) $\qquad \bar{r}_{i,j}, p_{i,j} = \Phi(s)_{i,j},$

(b) $\qquad v_{i,j}^0 = \bar{r}_{i,j}, \quad v_{i,j}^{(k)} = \max\left(v_{i,j}^{k-1}, \max_{(i',j')\in\mathcal{N}(i,j)}\left(\bar{r}_{i,j} + p_{i,j}(v_{i',j'}^{(k-1)} - \bar{r}_{i,j})\right)\right),$

(c) $\qquad \pi(s,(i_0,j_0)) = \underset{i',j'\in\mathcal{N}(i_0,j_0)}{\operatorname{argmax}} \, v_{i',j'}^K.$

Like VProp, MVProp is another implementation of value iteration with a deterministic mapping from the agent actions to position, however this time the model is constrained to propagate only positive rewards, since the propagation equation $\bar{r}_{i,j} + p_{i,j}(v^{(k-1)}_{i',j'} - \bar{r}_{i,j})$ can be rewritten as $p_{i,j}v^{(k-1)}_{i',j'} + \bar{r}_{i,j}(1 - p_{i,j})$ so the major difference with VProp is that MVProp focuses only on propagating positive rewards. Note therefore that all other remarks concerning more general versions of VProp also apply to MVProp.

## 4 EXPERIMENTS

We focus on evaluating our modules strictly using Reinforcement Learning, since we are interested in moving towards scenarios that better simulate tasks requiring interaction with the environment. For training, we employ an actor-critic architecture with experience replay. We collect transition traces of the form $(s^t, a^t, r^t, p^t, s^{t+1})$, where $s^t$ is the observation at time step $t$, $a^t$ is the action that was chosen, $p^t$ is the vector of probabilities of actions as given by the policy, and $r^t$ is the immediate reward. The architecture contains the policy $\pi_\theta$ described in the previous sections, together with a value function $V_w$, which takes the same input as the softmax layer of the policy, concatenated with the $3 \times 3$ neighborhood of the agent. $w$ and $\theta$ share all their weights until the end of the convolutional recurrence. At training time, given the stochastic policy at time step $t$ denoted by $\pi_{\theta^t}$, we sample a minibatch of $B$ transitions, denoted $\mathcal{B}$, uniformly at random from the last $L$ transitions, and perform gradient ascent over importance-weighted rewards:

$$\theta^{t+1} \leftarrow \theta^t + \eta \sum_{(s,a,r,p,s') \in \mathcal{B}} \min\left(\frac{\pi_{\theta^t}(s,a)}{p(a)}, C\right)\left(r + \mathbf{1}_{\{s' \neq \emptyset\}} \gamma V_{w^t}(s') - V_{w^t}(s)\right)\left(\nabla_{\theta^t} \log \pi_{\theta^t}(s,a)\right)$$
$$+ \lambda \sum_{(s,a,r,p,s') \in \mathcal{B}} \sum_{a'} p(a')\left(\nabla_{\theta^t} \log \pi_{\theta^t}(s,a')\right),$$
$$w^{t+1} \leftarrow w^t - \eta' \sum_{(s,a,r,p,s') \in \mathcal{B}} \min\left(\frac{\pi_{\theta^t}(s,a)}{p(a)}, C\right)\left(V_{w^t}(s) - r - \mathbf{1}_{\{s' \neq \emptyset\}} \gamma V_{w^t}(s')\right) \nabla_{w^t} V_{w^t}(s),$$

where $\mathbf{1}_{\{s' \neq \emptyset\}}$ is 1 if $s'$ is terminal and 0 otherwise. The capped importance weights $\min\left(\frac{\pi_{\theta^t}(s,a)}{p(a)}, C\right)$ are standard in off-policy policy gradient (Wang et al., 2016). The capping constant ($C = 10$ in our experiments) controls the variance of the gradients at the expense of some bias. The second term of the update acts as a regularizer and forces the current predictions to be close enough the the ones that were made by the older model. The learning rates $\eta$, $\lambda$ and $\eta'$ also control the relative weighting of the different objectives when the weights are shared.

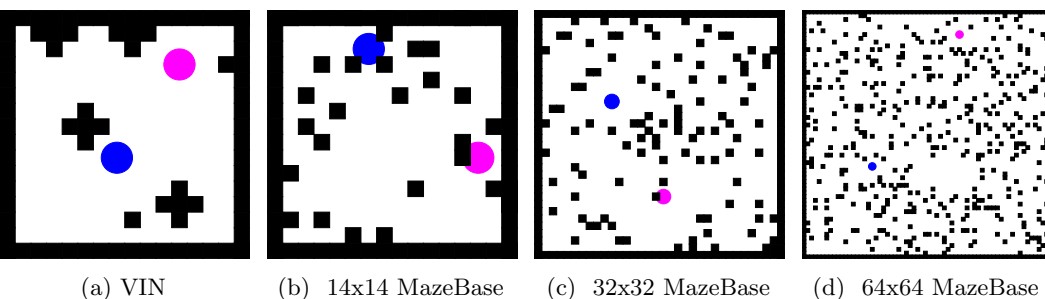

(a) VIN      (b) 14x14 MazeBase      (c) 32x32 MazeBase      (d) 64x64 MazeBase

Figure 1: Comparison between a random map of the VIN dataset, and a few random configuration of our training environment. In our custom grid-worlds, the number of blocks increases with size, but their percentage over the total available space is kept fixed. Agent and goal are shown as circles for better visualization, however they still occupy a single cell.

### 4.1 Grid-world setting

Our experimental setting consists of a 2D grid-world of fixed dimensions where all entities are sampled based on some fixed distribution (Figure 1). The agent is allowed to move in all 8 directions at each step, and a terminal state is reached when the agent either reaches the goal or hits one of the walls. We use MazeBase (Sukhbaatar et al., 2015) to generate the configurations of our world and the agent interface for both training and testing phases. Additionally we also evaluate our trained agents on maps uniformly sampled from the $16 \times 16$ dataset originally used by Tamar et al. (2016), so as to get a direct comparison with the previous work, and to confirm the quality of our baseline. We tested all the models on the other available datasets ($8 \times 8$ and $28 \times 28$) too, without seeing significant changes in relative performance, so they are omitted from our evaluation. We employ a curriculum where the average length of the optimal path from the starting agent position is bounded by some value which gradually increases after a few training episodes. This makes it more likely to encounter the goal at early stages of training, allowing for easier conditioning over the goal feature. Across all our tests on this setup, both VProp and MVProp greatly outperformed

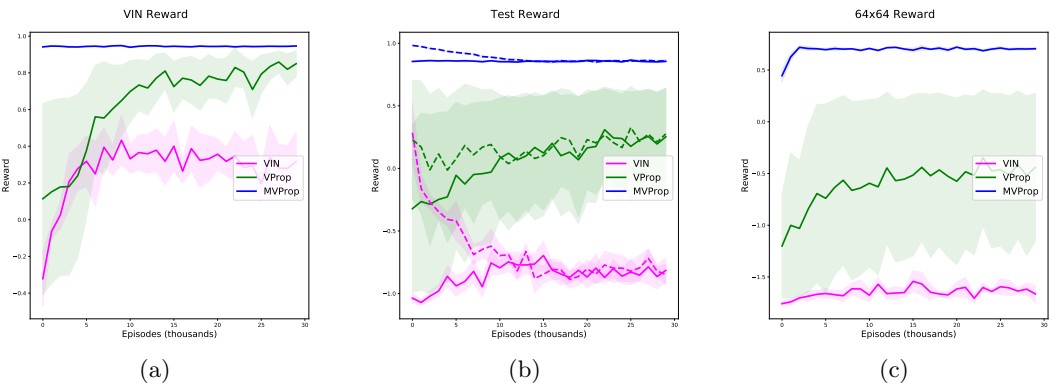

Figure 2: Average, min and max reward of all the models as they train on our curriculum. Note again that in the first two plots the the map size is $32 \times 32$. a and c demonstrate performances respectively on the VIN dataset and our generated $64 \times 64$ maps. b shows performance on evaluation maps constrained by the curriculum settings (segmented line), and without (continuous line).

our implementation of VIN. Figure 2 shows rewards obtained during training, averaged across 5 training runs seeded randomly. It's important to note that the original VIN architecture was mostly tested in a fully supervised setting (via imitation learning), where the best possible route was given to the network as target. In the appendix, however, Tamar et al. (2016) claim that VIN can perform in a RL setting, obtaining an 82.5% success rate, versus the 99.3% success rate of the supervised setting on a map of $16 \times 16$. The authors do not provide results for the larger $28 \times 28$ map dataset, nor do they provide learning curves and variance, however overall these results are consistent with the *best* performance we obtained from testing our implementation.

The final average performances of each model against the static-world experiments clearly demonstrate the strength of VProp and MVProp. In all the above experiments, both VProp and MVProp very quickly outperform the baseline. In particular MVProp very quickly learns a transition function over the MDP dynamics that is sharp enough to provide good values across even bigger sizes, hence obtaining near-optimal policies over all the sizes during the first thousand training episodes.

### 4.2 Tackling dynamic environments

To test the capability of our models to effectively learn non-static environments - that is, relatively more complex transition functions - we propose a set of experiments in which we allow our environment to spawn dynamic adversarial entities controlled by a set of fixed policies. These policies include a $\epsilon$-*noop* strategy, which makes the entity move in random

direction with probability $\epsilon$ or do nothing, a $\epsilon$-direction policy, which makes the entity move to a specific direction with probability $\epsilon$ or do nothing, and finally strictly adversarial policies that try to catch the agent before it can reach the goal. We use the first category of policies to augment our standard path-planning experiments, generating *enemies_only* environments where 20% of the space is occupied by agents with $\epsilon = 0.5$, and *mixed* environments with the same amount of entities, half consisting of fixed walls, and the remaining of agents with $\epsilon = 0.2$ The second type of policies is used to generate a deterministic but continuously changing environment which we call *avalanche*, in which the agent is tasked to reach the goal as quickly as possible while avoiding "falling" entities. Finally, we propose a third type of experiments where the latter fully adversarial policy is applied on 1 to 6 enemy entities depending on the size of the environment. These scenarios differ in difficulty, but all require the modules to both learn off extremely sparse positive rewards and deal with a more complex transition function, thus presenting a strong challenge for all the tested methods.

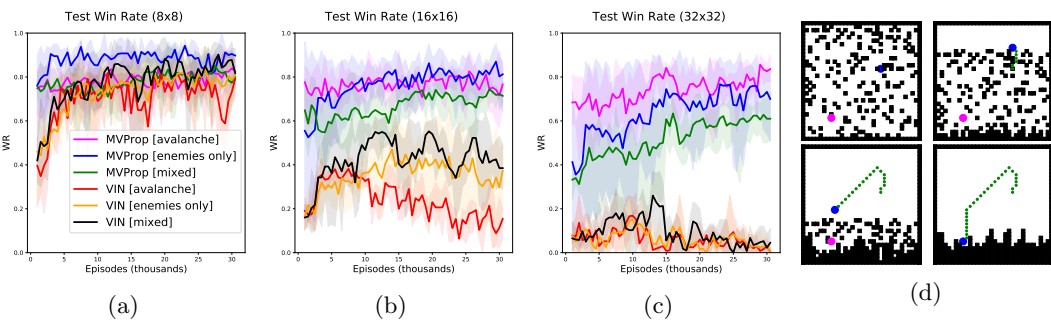

(a)   (b)   (c)   (d)

Figure 3: Average, min, and max, test win rate obtained on our dymamic experiments. Each agent was trained on the 8x8 instances of the scenario in a similar fashion to the static world experiments. Figure 3d shows an example of policy obtained after training on a avalanche testing configuration. Agent and goal are shown as circles for better visualization, however they still occupy a single cell.

As these new environments are not static, the agent needs to re-plan at every step, forcing us to train on 8x8 maps to reduce the time spent rolling-out the recurrent modules. This however allows us to train without curriculum, as the agent is already likely to successfully hit the goal in a smaller area with a stochastic policy. Figure 3 shows that MVProp learns to handle this new complexity in the dynamics, successfully generalising to 32x32 maps, much larger than the ones seen during training (Figure 3d), significantly outperforming the baseline.

## 4.3 STARCRAFT NAVIGATION

Finally, we evaluate VProp on a navigation task in *StarCraft: Brood War* where the navigation-related actions have low-level physical dynamics that affect the transition function. In StarCraft, it is common to want to plan a trajectory around enemy units, as these have auto-attack behaviours that will interfere or even destroy your own unit if found navigating too close to them. While planning trajectories of the size of a standard StarCraft map is outside of the scope of this work, this problem becomes already quite difficult when considering scenarios where enemies are close to the bottleneck areas, thus we can test our architecture on small maps to simulate instances of these scenarios.

We use TorchCraft (Synnaeve et al., 2016) to setup the environment and extract the position and type of units randomly spawned in the scenario. The state space is larger than the one used in our previous experiments and positive rewards might be extremely sparse, thus we employ a mixed curriculum to sample the units and their positions, allowing the models to observe positive rewards more often at the early stages of training and speed up training (note that this is also a requirement for the VIN baseline to have a chance at the task (Tamar et al., 2016)). As shown in Figure 4b, the MVProp is strong enough to plan around the low-level noise of the move actions, which enables the agent to navigate around the lethal enemies and

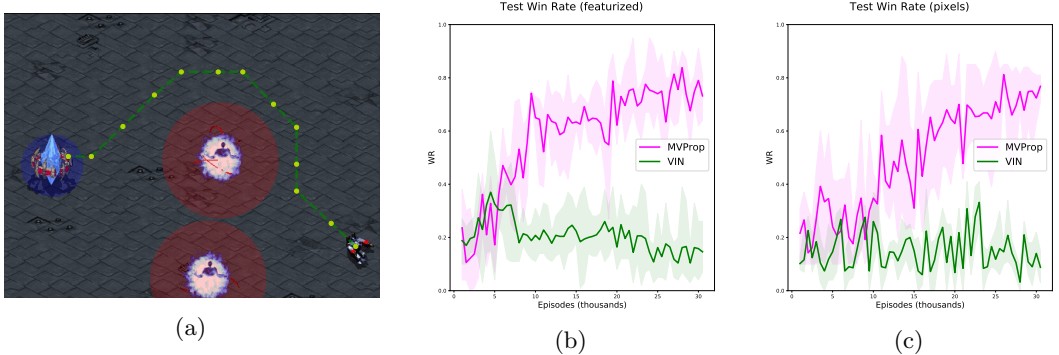

Figure 4: StarCraft navigation results. Figure 4a shows a generated trajectory on a random scenario at late stages of training. The red and blue overlays (not shown to the agent) indicate the distance required to interact with each enemy entity.

reach the goal on most random instances after training. Compared to the VIN baseline, the better sample efficiency translates into the ability to more accurately learn a model of the state-action transition function, which ultimately allows VProp to learn planning modules for this non-trivial environment. We also evaluate on the scenario directly from pixels, by adding two convolutional layers following by a max-pooling operator to provide capacity to learn the state features: as expected (Figure 4c) the architecture takes some more time to condition on the entities, but ultimately reaches similar final performances. In both cases, VIN struggles to condition correctly on the more complex transition function even with the curriculum providing early positive feedback.

## 5   Conclusions

Architectures that try to solve the large but structured space of navigation tasks have much to benefit from employing planners that can be learnt from data, however these need to be sample efficient to quickly adapt to local environment dynamics so that they can provide a flexible planning horizon without the need to collect new data. Our work shows that such planners can be successfully learnt via Reinforcement Learning when the dynamics of the task are taken into account, and that great generalization capabilities can be expected when these models are applied to 2D path-planning tasks. Furthermore, we have demonstrated that our methods can even generalize when the environment has dynamic, noisy, and adversarial elements, or with high-dimensional observation spaces, enabling them to be employed in relatively complex tasks. A major issue that still prevents these planners from being deployed on harder tasks is computational cost, since the depth increases with the length of the path that agents must solve, however architectures employing VI modules as low level planners have been successfully tackling complex interactive tasks (Section 1.1), thus we expect our methods to provide a way for such type of work to train end-to-end via reinforcement learning, even for pathfinding tasks found in different graph-like structures (for which we have at least the relevant convolutional operators). Finally, interesting venues where VProp and MVProp may be applied are mobile robotics and visual tracking (Lee et al., 2017; Bordallo et al., 2015), where our work could be used to learn arbitrary propagation functions, and model a wide range of potential functions.

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

## A    MORE GENERAL GRAPH STRUCTURES

VIN and VProp are, in their most general formulation, applicable to any graph-structured input. They ultimately belong to the general class of graph convolutional neural networks, and several variations of the value iteration modules specifically tailored to non-regular graph structures have been proposed (see e.g., Niu et al. (2017)). The three equations for VProp (Sections 3.2, 3.1) are applicable to any graph structure, as long as there is a one-to-one mapping between nodes in the neighborhood of the current position and actions (which is usually the case in navigation problems). Our work focuses on the simplest possible parametrization that is relevant in many navigation and pathfinding scenarios, while for more general graph structures, even assuming a deterministic model, the term $p_{i,j} v_{i',j'}^{(k-1)}$ should be extended into $p_{i,j,i',j'} v_{i',j'}^{(k-1)}$ to account for the edge weight between $i, j$ and $i', j'$. For regular graph structures such as 2D grids, this can be included in the embedding function $\Phi$, which outputs not just one parameter for each position but as many parameters as the neighborhood size. Other possibilities include using an attention mechanism for graph-convolutional neural networks in place of $p$ (see e.g., Tamar et al. (2016, section 4.4)). in such case, our method differs from the original VIN purely by the parametrization of the reward and the focus on the deterministic models, which we believe are relevant in navigation problems.

## B    AGENT SETUP

All the models and agent code was implemented in PyTorch (Paszke et al., 2017), and will be made available upon acceptance together with the environments. Most of the agents tested shared learning hyperparameters fitted to the VIN baseline to make comparison as fair as possible, and were validated using more than 5 random seeds across all our experiments. The second term in the $\phi^{t_1}$ update is supposed to play the role of TRPO-like regularization as implemented in Wang et al. (2016), where they use probabilities of an average model instead of the previous probabilities. We implemented an n-step memory replay buffer, observing that keeping only the last 50000 transitions (to avoid trying to fit predictions of a bad model) worked well on our tasks. In all our experiments we used RMSProp rather than plain SGD, with relative weights $\lambda = \eta = 100.0\eta'$. We also used a learning rate to 0.001 and mini-batch size of 128, with learning updates set at a frequency of 32 steps. We tested reasonable ranges for all these hyperparameters, but observed no relative significant changes when cross-validated over multiple seeds for most of them.

## C    MAZEBASE SETUP

The agents were tasked to navigate the maze as fast as possible, as total cost increased with time since *noop* actions were not allowed. Episodes terminated whenever the agent would take any illegal action such as hitting a wall, or when the maximum number of steps (set to roughly three times the length of the shortest path) would be reached. All entites were set with discrete collision boundaries corresponding to the featurised observation. The environments were also constrained so that only one entity could be present in any given cell at time $t$. Unless specified otherwise, attempting to walk into walls would yield a reward of $-1$, any valid movement would provide a reward of $-0.01 \times f(a_t, s_t)$, where $f(a, s)$ is the cost of moving in the direction specified by action $a$ in state $s$, and reaching the goal would give a positive reward of 1 to the agent. In our experiments we define $f$ as the L2 distance between the agent position at state $s_t$, and the one at $s_{t+1}$, to adjust for the real cost of moving diagonally.

For all static experiments, the ratio of un-walkable blocks over total space was fixed to 30%, and the blocks were sampled uniformly within the space, unless specified otherwise. This setting provided environments of decent difficulty, with chunky obstacles as well as harder, more narrow paths. Dynamic environments used different ratios of blocks (and other entities) vs walkable surface: *avalanche* environments filled between 20% and 30% of the surface with "falling" entities, *enemies only* spawned 10% (rounded to the bigger integer) of the surface with adversarial agents running A*, while *mixed* environments employed the same amount

| maps | VIN | VProp win rate | MVProp | VIN | VProp distance to optimal path | MVProp |
|---|---|---|---|---|---|---|
| $v$16x16 | $63.6\% \pm 13.2\%$ | $94.4\% \pm 5.6\%$ | 100% | $0.2 \pm 0.2$ | $0.2 \pm 0.2$ | $0.0 \pm .0$ |
| $32 \times 32$ | $15.6\% \pm 5.3\%$ | $68.8\% \pm 27.2\%$ | 100% | $0.8 \pm 0.3$ | $0.4 \pm 0.3$ | $0.0 \pm .0$ |
| $64 \times 64$ | $4.0\% \pm 4.1\%$ | $53.2\% \pm 31.8\%$ | 100% | $1.5 \pm 0.4$ | $0.5 \pm 0.4$ | $0.0 \pm .0$ |

Table 1: Average performance at the end of training of all tested models on the static grid-worlds with 90% confidence value, across 5 different training runs (with random seeding). $v$16x16 correspond to the maps sampled from VIN's 16x16 grid test dataset, while the rest of the maps are sampled uniformly from our generator using the same parameters employed at training time. The distance to the optimal path is averaged only for successful episodes.

of adversarial agents with the addition of static and stochastic entities (with and $\epsilon$-greedy stochastic policy) making up for 10% of the surface area, for a total of roughly 20% occupied blocks.

Our VIN models were based off the architecture employed by Tamar et al. (2016) in their grid-world experiments, which we used to also build VProp and MVProp models. The only significant difference between the models (beyond the number of maps used in the recurrency) consisted in VProp and MVProp using 8 input filters and unpadded convolutions. Note that we tested our VIN models using the same setup and saw no significant difference in these tests either.

## D  STARCRAFT SETUP

As TorchCraft by default runs at a very high framerate, we had to set the amount of skipped frames to 15, to allow us to be able to see relevant changes in the environment after each step; this is roughly double compared to other work done on the same platform (Usunier et al., 2016; Foerster et al., 2017). The rest of the environment parameters were kept to the default values. All entities were spawned in a similar fashion to the grid-world experiments, with an additional minimal constraint based on each entity's pixel size. At training time we employed a fixed max-distance curriculum to gradually increase the distance between spawned goal and agent, however we also prevented enemies from spawning during the first 500 episodes so as to allow the stochastic policy to quickly condition on the goal. Without this particular change in the curriculum setting we found learning to be generally more unstable, since the stochastic policy would need to naturally luck out. This could have been fixed by utilising a better exploration strategy, but we considered it outside the scope of this work.

We used a fixed 8x downsampling operator to reduce the size of the raw observation of 480x360 pixels to 60x45. Based on the experiment, this downsampled observation was then either fully featurised in a grid-world fashion, or transformed into greyscale (as it is typically done in other deep reinforcment learning experimental settings). We increased the capacity of the models by adding 2 additional convolutional layers with 32 filters and 7x7 and 5x5 kernels, and max-pooling layers with stride and extent equal to 2 to further reduce the dimensionality before the recurrent step.

