# OpenReview forum: "Value Propagation Networks"
_ICLR.cc/2019/Conference_

### Official Review · AnonReviewer3 · 2018-11-02

**Rating:** 7
**Confidence:** 3

**Review:**

Update:

I thank the authors for the response. Unfortunately, the response does not mention modifications made to the paper according to the comments. According to pdfdiff, modifications to the paper are very minor, and none of my comments are addressed in the paper. I think the paper shows good results, but it could very much benefit from improved presentation and evaluation. I do recommend acceptance, but if the authors put more work in improving the paper, it could have a larger impact.

------

The paper proposes a learnable planning model based on value iteration. The proposed methods can be seen as modifications of Value Iteration Networks (VIN), with some improvements aimed at improving sample efficiency and generalization to large environment sizes. The method is validated on gridworld-type environments, as well as on a more complex StarCraft-based domain with raw pixel input.

Pros:
1) The topic of the paper is interesting: combining the advantages of learning and planning seems like a promising direction to achieving adaptive and generalizable systems.
2) The presentation is quite good, although some details are missing.
3) The proposed method can be effectively trained with reinforcement learning and generalizes well to much larger environments than trained on. It beats vanilla VIN by a large margin. The MVProp variant of the method is especially successful.

Cons:
1) I would like to see a more complete discussion of the MVProp method. Propagation of only positive rewards seems like somewhat of a hack. Is this a general solution or is it only applicable to gridworld navigation-type tasks? Why? If not, is the area of applicability of MVProp different from VProp? Also, is the area of applicability of VProp different from VIN? It’s important to discuss this in detail.
2) I wonder how would the method behave in more realistic gridworld environments, for instance similar in layout to those used in RL navigation literature (DMLab, ViZDoom, MINOS, etc). The presented environments are quite artificial and seem to basically only require “obstacle avoidance”, not so much deliberate long-distance planning.
3) Some details are missing. For instance, I was not able to find the exact network architectures used in different tasks.
Related to this, I was confused by the phrase “As these new environments are not static, the agent needs to re-plan at every step, forcing us to train on 8x8 maps to reduce the time spent rolling-out the recurrent modules.” I might be misunderstanding something, but is there any recurrent network in VProp? Isn’t it just predicting the parameters once and then rolling our value iteration forward without any learning? Is this so time-consuming?
4) Why does the performance even of the best method not reach 100% even in the simpler environments in Figure 2? Why is the performance plateauing far from 100% in the more difficult case? It would be interesting to see more analysis of how the method works, when it fails, and which parts still need improvement. On a related topic, it would be good to see more qualitative results both in MazeBaze and StarCraft - in the form of images or videos.
5) Novelty is somewhat limited: the method is conceptually similar to VIN.

To conclude, I think the paper is interesting and the proposed method seems to perform well in the tested environments. I am quite positive about the paper, and I will gladly raise the rating if my questions are addressed satisfactorily.

---

> ### Author Response · Authors · 2018-11-26
> **Rebuttal to AnonReviewer3**
>
> We thank the reviewer for the suggestions and positive comments. We would like to answer some of their questions:
>
> 1) This is an interesting question, thank you for asking. In general MVProp is applicable to any path-planning task where the dynamics are deterministic, the agent is generally interested in finding the path with lowest cost (or, in this case, highest reward), and the reward function associated to the task does correspond to simply in one based on path-planning (i.e. some cost is associated to moving and getting into invalid states, and some reward is given for reaching goals). This applies to a variety of path-planning problems, but if the environment has a more nuanced reward function that would provide quicker feedback from minimising negative cost, VProp might learn faster. Our experiments show that our models work outside of the deterministic assumption (see experiments on stochastic environments in Section 4.2).
>
> 2) We are not certain it is clear we would gain further intuition from looking at DMLab, VizDoom, or other maze environments, since a birds-eye version of the tasks available in these environments would be relatively similar to either our static grid-world setup or the dynamic ones, but much slower and resource intensive. That said, while we might indeed explore in the future such tasks, we feel that VProp and MVProp might be best used as planning modules within a larger and more complex planning architecture that we mentioned in Section 1.1, such as Niu et al. (2017), and Gupta et al. (2017). Also note that these environments provide a standard setup (i.e. first person view, high degree of partial observability, some stochasticity) that is definitely beyond the scope of all our models and baselines.
>
> 3) At each agent step, our model (and the baseline) needs to convolve for K steps, where K is equal to roughly the distance to the goal (step-wise) for each step. If the environment is deterministic and “static”, i.e. goal state and unreachable states do not change, the agent only needs to do this process once, however if the environment changes stochastically the model assumptions are broken, thus we need to at least allow to replan after some amount of steps (and we chose 1 to give the VIN baseline a fair chance). There are ways to improve upon this, such as building a hierarchical planner directly inside the VProp modules, but such methods require significant changes and are pretty much future work.
>
> 4) Figure 1 shows the (average) final reward obtained in the static settings, where the optimal reward is $goal_reward - optimal_steps * step_cost$, where reward_goal = 1 and cost_step = 0.01. MVProp very quickly reaches optimality, which on average gives reward slightly below 1 in all test environments, while VProp is more unstable when generalising to larger instances. Please see Table 1 in the Appendix for more precise numbers. In terms of videos, we will show a demo of the models working at the conference.
>
> 5) We don’t think of our work as necessary just an extension of VIN, but more of a principled way to learn planning modules that are fully convolutional and that can generalize across wildly different planning horizons and input sizes while interacting with the environment (thus via RL). The great majority of agents / planners based on deep neural networks tends to either ignore the problem altogether, or use fixed transformations on the input, leading to resolution and/or information loss. We look forward to seeing more work tackling this problem, and we hope VProp and MVProp will provide a good step in that direction.

---

### Official Review · AnonReviewer2 · 2018-11-03
**Missing information in the exposition**

**Rating:** 6
**Confidence:** 3

**Review:**

Update:
I thank the authors for their clarifications. I have raised my rating, however I believe the exposition of the paper should be improved and some of their responses should be integrated to the main text.

The paper proposes two new modules to overcome some limitations of VIN, but the additional or alternative hypotheses used compared to VIN are not clearly stated and explained in my opinion.

    Pros :
    - experiments are numerous and advanced
    - transition probabilities are not transition-invariant compared to VIN
    - do not need pretraining trajectories

    Cons :
    - limitation and hypotheses are not very explicit

    Questions/remarks :
    - d_{rew} is not defined
    - the shared weights should be explained in more details
    - sometimes \psi(s) is written as parametrized by \theta, sometime not
    - is it normal that the \gamma never appears in your formula to update the \theta and w? yet reading the background part I feel that you optimize the discounted sum of the rewards, is it the case?
    - I think there is a mistake in the definition of 1_{s' \neq \emptyset }, it is 1 if s' is NOT terminal and 0 otherwise, am I wrong?
    - why do you need the parameters w to represent the value function V, if you already have v^k_{i,j} available? is it just to say that your NN is updated with two distinct cost functions?
    - I did not understand the assumptions made by VProp, do you consider that the transition function T is known? this seems to be the case when you explain that transitions are deterministic and that there is a mapping between the actions and the positions, but is never really said
    - Compared to VIN, VProp uses an extra maximum to compute v^k_{i, j}, why? In this case, the approximation of the value function can never decrease.
    - How is R_{a, i, j, i ', j'} broken into r^{in}_{i ', j'} - r^{out}_{i, j} in VProp? Is the reward function known to the agent at all points?
    - In MVProp, can r_{i, j} be negative?
    - In MVProp, how does the rewriting in p * v + r * (1-p) shows that only positive rewards are propagated? Does not it come only from the max?
    - In the experiments, S is not fully described, \phi(s) neither

---

> ### Author Response · Authors · 2018-11-26
> **Rebuttal to AnonReviewer2**
>
> We thank the reviewer for the positive comments. Here’s some answers which will hopefully clarify some of the questions posed:
>
> >d_{rew} is not defined
>
> Regarding \drew, in the case of VIN we do implicitly refer to it with “output channels”, as it really is just some variable defining the number of channels used for the embedding function. We have adjusted Section 2.2 to make this more evident. Note also that in the case of VProp, it is 3 (r^in, r^out, p), and 2 (r, p) for MVProp.
>
> >the shared weights should be explained in more details
>
> When we say “shared weights” we mean it literally: the recurrence step is done by the same network layers, as opposed to convolving at each step using different parameters. That (trivially) reduces the amount of parameters needed when the network is fully unrolled, and it allows us to generalise to larger environments by unrolling more.
>
> >Inconsistent use of theta in \psi [sic.], missing gamma, and definition of 1_{s' \neq \emptyset }
>
> We are not using \psi anywhere. If instead you are referring to \Phi, for consistency we have removed the theta from previous equations prior to Section 4 in the new revision.
>
> Indeed, there should be a \gamma behind V_{w^t} for both updates, and there's a missing "not" in the definition of 1_{s' \neq \emptyset }, thank you for noticing both. We have fixed them in the new revision.
>
> >why do you need the parameters w to represent the value function V, if you already have v^k_{i,j} available? is it just to say that your NN is updated with two distinct cost functions?
>
> Exactly, the loss in off-policy actor-critic for the value head is indeed different from the one applied to the policy head, and the parameters updated are not the same, so we felt that it was clearer to split the two. Furthermore, to be completely clear, V^k{i, j} is the value function inside the planning module, while V is the overall value function of the final “policy” layer used within actor-critic.
>
> >I did not understand the assumptions made by VProp, do you consider that the transition function T is known? this seems to be the case when you explain that transitions are deterministic and that there is a mapping between the actions and the positions, but is never really said
>
> We do not assume we know the transition function T, and in fact we do learn some parameters of it. However we do assume that the function is constrained in certain ways, i.e. at most one state may be accessible from another state, given an action.
>
> >Compared to VIN, VProp uses an extra maximum to compute v^k_{i, j}, why? In this case, the approximation of the value function can never decrease.
>
> The extra “max” is equivalent to adding an extra action of staying in the same state with zero immediate reward. We would not lose any generality if immediate rewards are always positive. In our case, it is a convenient way to represent absorbing states (i.e., goal states).
>
> >How is R_{a, i, j, i ', j'} broken into r^{in}_{i ', j'} - r^{out}_{i, j} in VProp? Is the reward function known to the agent at all points?
>
> The reward broken down in two values is a choice of parametrization. This is an assumption, which drastically reduces the number of parameters to learn (because there are only two values per state instead of R(i,j,a,i’,j’) ). Our point here is to say that this parametrization is sufficient to represent cases of interest, but in general this is an assumption that does not always hold.
>
> >In MVProp, can r_{i, j} be negative?
>
> It could (in which case it would decrease the values of nearby states that are propagated to the current state). In practice, stopping the propagation can be carried out by near-0 values of the propagation gate, and rewards are constrained to be positive by a sigmoid activation function on the reward channel.
>
> >In MVProp, how does the rewriting in p * v + r * (1-p) shows that only positive rewards are propagated? Does not it come only from the max?
>
> Yes, the max with the current value is what makes negative values of the reward in a state not propagate to nearby states (because a high value of r_{i,j} propagates to nearby states by first increasing v_{i,j}, which itself is propagated to the neighboring v_{i’,j’} at further iterations. Negative values of r_{i,j} would not update v_{i,j} and thus wouldn’t be propagated to nearby states. We believed the rewriting as pv + r(1-p) helps understanding that negative values of r_{i,j} will not be used to update v_{i,j} in the first iteration. We will clarify this statement.
>
> >In the experiments, S is not fully described, \phi(s) neither
>
> For the grid-world experiments, the state is described in Section 2.2, and \phi(s) is a fixed function that splits each feature into its own channel; for the StarCraft experiments, the grid-world featurization is similarly done (and based on TorchCraft’s API), while in pixel space \phi(s) is added to the network as two extra convolutional -> max pooling layers, as described in Appendix D.

---

### Official Review · AnonReviewer1 · 2018-11-06
**Interesting extension of the original value iteration networks (VIN), promising work**

**Rating:** 7
**Confidence:** 3

**Review:**

The paper presents an extension of the original value iteration networks (VIN) by considering state-dependent transition function, which alleviates the limitation of VIN to translation-invariant transition functions and further constraining the reward function parametrization to improve sample efficiency of learning to plan algorithms. The first problem is addressed  by interpreting transition probabilities as state-dependent discount factors, given by a sigmoid function that takes as input state features. The second problem is addressed by defining the reward function as the difference between an input reward and an output cost. Obstacle states are given a high cost. The proposed method is evaluated on random grids of different sizes, of the same type as the grids considered in the VIN paper. Comparaisons with VIN show that the proposed MVProp approach outperforms VIN by several orders of magnitude and can learn optimal plans in less than a thousand episodes, compared to VIN that doesn't seem here to learn much even after 30 thousands episodes.
The paper is well-written in general. Certain aspects of value iteration networks were explained too briefly and the reviewer had to re-read the original VIN paper to grasp certain details of the proposed approach. This work is an interesting improvement of VIN, but somehow incremental in nature as the improvement is limited to slightly changing the reward and transition representations. However, the resulting performance seems very impressive, especially for larger grids. One question that needs to be clarified is: how is this work situated with respect to the body of work on RL? How does this method compare empirically to model-free algorithms such as DDPG and PPO?

---

> ### Author Response · Authors · 2018-11-26
> **Rebuttal to AnonReviewer1**
>
> We thank the reviewer for the very positive comments. Regarding their point about comparison against standard model-free algorithms, we didn’t compare against them because these agents can’t be applied well to the experimental setup. More precisely:
>
> - typical models used with these algorithms cannot deal with varying input sizes, unless you engineer a function that would downsample / upsample the observation to a particular shape. One could learn in principle such a function, but that would require an entirely different experimental and evaluation loop, e.g. one that would keep the “core” agent model frozen while training only an embedding function (which would however likely result in unstable learning). Making standard model size-invariant w.r.t. the observation size is a problem that we have decided to tackle by employing fully convolutional models, but this differs from most DRL work.
>
> - the models would need to be much bigger in terms of hyperparameters, and would most likely need a better curriculum to deal with the sparse positive signals, thus also making comparison trickier.
>
> Tamar et al. (2016) (whose experimental setup is similar to ours) show these points pretty clearly.
>
> Please also note that we could replace our actor-critic update rule (and agent setup) with PPO (and DDPG if we were to use a continuous action space), but the focus of our paper was the planning module, so all our experiments share the same agent setup.

---

### Meta-Review · Area_Chair1 · 2018-12-14
**Interesting idea, reviewers were positive and indicated presentation should be improved.**

**Confidence:** 3
**Recommendation:** Accept (Poster)

**Metareview:**


Interesting idea, reviewers were positive and indicated presentation should be improved.